# Daoist Reflections on the See-Saw of Contingency and Autonomy: The *Laozi* and *Zhuangzi* in Dialogue with Sandel, Rosa, Rorty, Gray

Paul Joseph D'Ambrosio

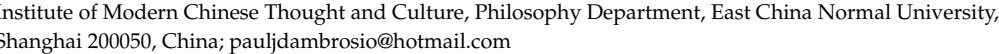

Institute of Modern Chinese Thought and Culture, Philosophy Department, East China Normal University, Shanghai 200050, China; pauljdambrosio@hotmail.com

**Abstract:** Nearly all philosophical inquiry is rooted in contingency. From ontology and theories of God to politics and ethics, dealing with, explaining, planning for, or even following contingency is a consistent theme. In the background of their recent works, Michael Sandel, Hartmut Rosa, John Gray, and Richard Rorty all see contingency and autonomy in a see-saw relationship: more of one correspondingly results in less of the other. Daoist philosophical reflections provide a different take on contingency. We can still have an experience of "self" and of making choices without positing any notion of autonomy outside of contingency.

**Keywords:** contingency; autonomy; Daoism; *Zhuangzi*; *Laozi*; Michael Sandel; Hartmut Rosa; Richard Rorty; John Gray

## 1. Introduction: The Importance of Contingency

Contingency is at the root of much philosophical reflection. Depending on how it is theorized, it contributes to the foundational problems in everything from ontology and theories of God to politics and ethics. Planning for, dealing with, explaining, or even following contingent factors in life is at the core of many of these discussions.

For many, contingency is something to be overcome. One route is explanation. Karma promises a final balance. Divine grace, God's plan, or simply 'everything happens for a reason' are other approaches to maintain sanity when faced with the unpredictable. (Darwinism, especially in social settings, is often cheaply referred to as a necessary principle which overrides contingent factors in life. Darwin was, however, only highlighting the very opposite.) Even with these justifications, however, we do not always feel satisfied. More proactive attempts to level the playing field are supposed to supply comfort. We have various techniques to defend ourselves against genetic anomalies, cataphoric weather, bad people, bad governments, or just plain bad luck. Though we may never escape their grasp, we do our best to plan against and deal with the fickle favoritism of chance, fate, or pure coincidence.

To some degree, the capriciousness of life is simply something to accept. Aristotle thought, for example, that there is only so much one can do to guarantee their own happiness. The virtue of one's family members or the political system one is born into, for example, are to some degree beyond their control. And if disasters strike, if tragedies take place, or simply if bad things happen, there may very well be little one can do. Even if one cultivates virtue, is scrupulous in making good friends, and generally commits to securing as much of life as possible, there is still much that is ultimately outside of one's control.

In addition to rather hard-hitting examples of contingency in 2020, we also have the publication of three titles theorizing about various dimensions of contingency. Michael Sandel's *The Tyranny of Merit*, Hartmut Rosa's *The Uncontrollability of the World*, and John Gray's *Feline Philosophy*, each contain arguments about the role of contingency today. However, in none of these books do the COVID-19 virus, Black Lives Matter movements, or

climate crisis figure significantly.[1] Their approaches are chiefly concerned with underlying conceptions of luck, chance, predictability, control, perfection, rationality, and autonomy. The implications of their respective discussions are immense. They claim to have the potential to impact how we think about politics and justice, technology and our relationship to the natural and social world, as well as the very meaning of life.

In many ways, the approaches that Sandel, Rosa, and Gray utilize to discuss their respective issues—especially when looked at with an eye for contingency—share a common theme. They all see contingency and autonomy in a see-saw relationship. More of one correspondingly results in less the other. Consequently, while Sandel and Rosa grant that the natural world, society, and even one's body or perhaps "self" is largely comprised of contingent factors, they reserve a separate space for each person's autonomy. In other words, the individual is, to whatever degree and in whatever dimension, not entirely at the mercy of contingency. The same is true for Richard Rorty, whose *Contingency, Irony, and Solidarity* is sometimes (mis)taken to be a celebration of relativistic "blowing in the wind".

This paper will outline the work of Sandel, Rosa, and Rorty, demonstrating that their understandings of autonomy leave the individual outside the direct influence of contingent factors and thereby somehow abstracted from the concrete world. While Gray backs away from assigning autonomy to the individual, he does not provide much in the way of outlining what this might look like in a constructive way. These views will be contrasted with classical Daoist philosophy as found in the *Laozi* 老子 (Lao-tzu or Tao-Te-Ching) and *Zhuangzi* 莊子 (Chuang-Tzu) as well as the later developments by Guo Xiang 郭象 (Kuo Hsiang) (d. 312). Daoist thought offers philosophical reflections whereby not only are natural, socio-political, and bodily factors all subject to contingency, but one's self[2]— whatever that may be—is completely contingent as well. Beyond simply appreciating the thoroughgoing nature of contingency Daoism also offers "self-so", "non-action", and "knowing contentment" as productive outlets for conceiving of human effort and conscious action. This model allows for a complete reframing of how we perceive of the see-saw relationship between contingency and autonomy, and is especially impactful for how we think of ourselves. Accordingly, what Sandel, Rosa, Rorty, and Gray take as a necessary balancing act between autonomy and contingency is reconceptualized so that anything resembling autonomy only exists within complete contingency.

## 2. Sandel: Meritocratic Hubris

Michael Sandel exploded his professional academic career with the publication of *Liberalism and the Limits of Justice* (1982). It is a treatise against the work of John Rawls, mainly criticizing the foundation of Rawls' notion of justice: the abstract account of the individual. Sandel insists on including 'encumberedness'[3] that is, recognizing the importance of concrete factors, in theorizing about both the individual and justice. This has led many to employ the label "communitarian" in describing Sandel, and (even more) critically, charge him with promoting a "merely situated self". In other words, he is accused of presenting a self that is constituted entirely through circumstance; with no sense of person, will, or autonomous power outside social roles, relationships, and the pure happenstance in which it finds itself in in terms of its own qualities as well as its socio-political and natural environments. Defending himself against such a charge Sandel says his is a conception of "a self that is situated, but reflectively situated". (Sandel 2016) One's "constitute aims and attachments" (Sandel 2005, p. 167) are influential, but they are not everything. The "enduring attachments and commitments that, taken together, partly define the person I am" (Sandel 1982, p. 167) should be recognized as such. There is, however, still always some "I" behind it all. Unable to clarify exactly what it is, Sandel provides a general description: "As a self-interpreting being, I am able to reflect on my history and in this sense to distance myself from it, but the distance is always precarious and provisional, the point of reflection never finally secured outside the history itself" (Sandel 1982, p. 179).

In his 2020 *The Tyranny of Merit: What's Become of the Common Good?* Sandel applies his conception of the person and the visions of justice this entails to contemporary social

and political life in America. The guiding mantra in the U.S. is "Those who work hard and play by the rules should be able to rise as far as their talents will take them". Manifest in numerous ways, we find this idea undercutting everything from college admissions and CEO pay to discussions of welfare, insurance, and nearly every facet of the free market. It ignores, or rather selfishly structures, how we think about what we owe one another. The common good is defined negatively: not getting in the way of others or impeding their rights. This comes, however, at a cost. If I should stay out of your way (and that might be the key to your success) then I have no obligations to impact you positively either—and vice versa. As citizens we owe each other nothing more than yielding and respecting the right of way. Making a positive contribution is not a necessary (and often not even desirable) component of this ethos.

Merit has come to fill the moral void left in the wake of social, political, and ethical individualism. The promise is that whatever one gets within a meritocratic system is what they deserve. Those who succeed, be it with degrees, titles, fame, money, or even respect, reap the rewards they are owed. Anyone who comes up short is similarly deserving of their own failures. Winners work harder, are more talented, smarter, or just plain better. That is why they have won. They are justified in viewing themselves this way. In addition to material goods, they should enjoy the esteem, honor, and recognition society bestows. For losers, the same logic must apply. They do not work as hard, they lack talent, are not smart enough, or just do not cut it. This means not only lacking material goods, perhaps even necessary ones, and shouldering social censure and humiliation but also thinking of oneself in that way as well. They lose because they deserve to lose—and they should know that.[4]

The problem is thereby structural. It is not just that our systems of merit is skewed towards certain genders, races, economic backgrounds, and other discriminatory practices. To be sure, institutional defects are undeniably a bad thing. They should be dealt with, thoroughly and promptly. However, Sandel's analysis calls not just for adjusting the meritocratic knobs to better control for the real winners and losers. He criticizes that such stringent calculations of moral desert isolate individuals, and in doing so correspondingly eschew considerations for others.

> "[F]or the more we think of ourselves as self-made and self-sufficient, the harder it is to learn gratitude and humility. And without these sentiments, it is hard to care for the common good". (Sandel 2020, p. 14)

Specifically, we need to rethink our relationship to our own skills, talents, and perhaps even motivation. They are not a matter of one's own doing, "but a matter of good luck" (Sandel 2020, p. 122). We do not choose our natural capacities anymore than we choose our own DNA. To think of ourselves as deserving of the rewards our talents reap is akin to assigning value to genetics.[5] The same is true for how we think of others. "[T]hat I live in a society that prizes the talents I happen to have is also not something for which I can claim credit" (Sandel 2020, p. 123). The fact that certain skills or particular aptitudes are lavishly praised while others are moderately ignored can be, to whatever degree, understood as an accident of history and culture. Here comparing social media influencers with . . . well maybe most other jobs, is a great example. To put it differently, not everyone who works hard and plays by the rules gets into the NBA. It is, to a large extent, a genetic lottery. And the fact that the NBA exists, and the amount of money surrounding it, are hardly representative of necessary (or even ideal) moral outcomes.

Our way out (and to some extent 'back') is to recognize the importance of contingency in these areas. Sandel ends his book with a quote he mentions a number of times throughout: "There, but for the grace of God, or the accident of birth, or the mystery of fate, go I" (Sandel 2020, p. 227). We prevent winners from 'inhaling too deeply of their success' and evade the humiliation and resentment felt by losers through recognizing that the talents people have or lack and how they are either esteemed or slated are all accidental and to some degree even mysterious.

Meritocratic hubris is the problem: "The more we view ourselves as self-made and self-sufficient, the less likely we are to care for the fate of those less fortunate than ourselves" (Sandel 2020, p. 59) and "the less reason we have to feel indebted or grateful for our success" (Sandel 2020, p. 42). Appreciating contingency is the first step toward a way out.

### 3. Rosa: Uncontrollability

Hartmut Rosa's sociological works track differences between pre-modernity, modernity, and post-modernity as categorical advances in the degree of acceleration found in social systems and individuals. The transition from pre-modernity to modernity is marked by "dynamic stabilization". In pre-modern societies most things are relatively fixed. Institutional and even individual stability is achieved through the steady reproduction of norms. Modern societies require increase just to stay the same. Today nearly every social system and even each person must constantly accelerate just to stay in place. The economy is stable when it grows, a stagnant housing market is a declining one, and only a professor whose CV is constantly updated remains relevant (or even employable).

Behind acceleration is the drive to make the world more accessible, available, and attainable. Nearly all technology speeds up the amount or number of activities per unit of time. Cars and airplanes move us faster, while washing machines and cell phones allow for multitasking. The goal is to increasingly obtain more experience or control of the world. Rosa outlines the promise of modernity: Once we have established a significant grasp on X, when things are moving efficiently enough, we will have the free time to do the things we really want—to finally feel in touch with others, the world, and ourselves. Various types of unpredictability and uncontrollability are among the major obstacles.

The possibility of fulfilling promises made by acceleration is undercut by a fundamental paradox: acceleration only begets acceleration. When we use the washing machine, we have not suddenly freed ourselves up; instead, we are now able to, and further feel the need to, have other things to do. This logic pervades nearly all areas of life. The speeding up of the world only increases its speed and this is reflected within technologies, social systems, and existential experiences—it further characterizes their relations as well.[6] Relatedly, increasing predictability and controllability only broaden our distance from others, the world, and even ourselves. As acceleration expands our ever-encompassing grasp makes the world ever more knowable, predictable, and controllable. Life becomes progressively mechanistic. And we become alienated.

Alienation, a widely discussed and notoriously obtuse topic, is succinctly described by Rosa as "the sense of a mute, cold, rigid, or failed relationship to the world". (Rosa 2019, p. 35) Senses of community, including political participation, concern for family, friends, neighbors, and even the self are lessened as individualism, (over) rationalization, and the accompanying sterilization of interaction rise. Membership to groups becomes goal-driven as opposed to being resources for developing co-relational identity. Even 'transcendent' spheres, such as art, nature, history, the divine, and time, lose their potency. They become less thickly interwoven in our lives. Reducing otherwise complex experiences to calculations, heavy reliance on robust institutionalization, and increasing reach of rating and ranking systems leaves us in a world of dull echoes. More and more of the unpredictability and uncontrollability of the world is crowded out when technology, coupled with our own standardized expectations, allow only the least possible amount of contingency. The very potential for the world to speak to us is thus muted through instrumentalized relations.

Silencing the world means no longer resonating with it. And as we thus crowd out contingency through attempts at mastery, we correspondingly push away the possibility to be touched by the world and by others. The subject is no longer transformed by the world and the world is no longer transformed by the subject. Both operate upon one another, but neither enter an engagement where both sides can be heard. Unpredictable, uncontrollable, and un-engineerable change cannot, and in some sense *should not*, occur. Modern subjects feel this loss of resonance in a variety of ways. "People who are unhappy or depressed are likely to perceive the world as 'bleak, empty, hostile, and colourless', while experiencing

themselves and their inner world as 'cold, dead, numb, and deaf'".[7] In such cases, "[t]he axes of resonance between self and world remain silent" (Susen 2020, p. 316). When we relate to the world as composed of things to be predicted and controlled, we are not affected and meaning cannot be co-constituted. Everything—in some cases even our own bodies and selves—become dead objects. Resonance is not possible, all things either fit rigid expectations or else must be re-managed or ignored. The alternative involves reimagining the world and our relations to it:

> If we no longer saw the world as a point of aggression, but as a point of resonance that we approach, not with an aim of appropriating, dominating, and *controlling* it but with an attitude of listening and responding, an attitude oriented toward self-efficacious adaptive transformation, toward mutually *responsive reachability*, modernity's escalatory game would become meaningless and, more importantly, would be deprived of the psychological energy that drives it. (Rosa 2020, p. 108)

Opening to the possibility of resonance means opening to the possibility that there are other voices, that they have something to say, that what they say is uncontrollable, unpredictable, un-engineerable. These voices might affect one in equally uncontrollable, unpredictable, and un-engineerable ways, and one's own voice has the same potential. Neither side nor the change brought about can be managed or mastered. "Resonance always has the character of a *gift*". (Rosa 2020, p. 59) Although we do not know what the gift might be and cannot even be sure we want it.

As the world becomes more available, accessible, and attainable in increasingly predictable and controllable ways our expectations for how we will be affected become stiffer. Many areas of life are now dominated by rigid regulations backed, institutionally and psychologically, by an instrumentalizing logic of sameness.[8] Reducing complex environments to their barest subject–object networks, we increasingly expect not only to "plug and play" with various components, but also that ways in which concrete subjects are affected also be the same. The growing reach of institutionalization in various spheres tends towards turning social interactions into production-line-like processes. Inputs and outputs should be the same—even when they involve, on both sides of the equation, actual persons. Respecting the (relatively unique and diverse) autonomy of each side is a thoroughgoing requirement for inviting an openness to resonance.

Given contemporary social structures and mentalities an orientation towards an openness for resonance sounds excessively passive. However, what might seem as an extreme really only appears as such due to an overemphasis on the opposite today. Even if we allow some degree of uncontrollability, unpredictability, and un-engineerability in the world and in others we are well away from making a significant shift. One must also recognize oneself in this way. Much of what constitutes our 'self' is already extremely contingent. In agreement with consensuses across many disciplines, Rosa reminds:

> Whom we erotically or sexually desire, what arouses or attracts us is something beyond our own volition. Likewise, our taste in food and drink, music and literature, clothing and fashion, where and how we want to live, along with our wants, our longings, our sympathies and antipathies are to a large extent simply 'given' to us. (Rosa 2020, p. 102)

Moving away from excessive rationalization, instrumentalization, and the desire to control, predict, and manage what happens, begins with not treating one's own self in this way. Much of what makes up one's personality must already be taken as unpredictable and even uncontrollable. In other words, the various dimensions of one's own self are already a starting point for resonance. One can begin by resonating with themselves.

In this relationship, as with any other, the subject cannot be passive. Mere acceptance cannot characterize "a form of world-relation, in which subject and world meet and transform each other" (Rosa 2016, p. 298). Again, this applies just as much to the world as it does to one's own self. Our attitude is important. How we conceive of our autonomy is critical. The right limits must be in place, and the right content must fill those constraints.

We do not bring about resonance by striving too hard—especially as this often transforms into some form of instrumentalization—but we do need purposeful action. We do need to affirm and own our own voice:

> If we were to *always give in* to our desires, we would be nothing but pure "voluptuaries" (or "wantons") [ . . . ]. We would cease to be sane, accountable subjects and would be incapable of living a life. We would lack our "own voice" with which to respond to our desires. (Rosa 2020, p. 103)

Borrowing heavily from Charles Taylor, Rosa introduces the importance of "strong evaluations". The subject is still expected to maintain a distance from their desires, likes and dislikes, and many other "given" aspects. These include, as mentioned above "our taste in food and drink, music and literature, clothing and fashion, where and how we want to live, along with our wants, our longings, our sympathies and antipathies" (Rosa 2020, p. 102).[9] Like Sandel who defends against critics who accuse him of promoting a "merely situated self" Rosa must maintain that his position does not lead to becoming "pure voluptuaries". In order to reintroduce resonance contingency must be recognized all around, but there is still a subject to reflect, react, and rethink their own role.

Both Sandel and Rosa seek an alternative to what they view as over identifying with contingent factors—they are cautious about slipping into accusations of advocating a self that simply blows with the wind. However, swaying too far the other way, i.e., too much abstracted autonomy and individualism leads to a corresponding lack of appreciation of their being some degree of "chance" and their "uncontrollability". They must seek a middle ground where some sense of self is maintained, and one strong enough to be responsible for choosing and evaluating our likes, dislikes, ends, and values. Sandel respects chance and wants us to correct our meritocratic system to reflect its pervasiveness. But he also reminds us that the self is always capable of reflecting on its own conditions. Rosa too thinks we have enough autonomy to evaluate our own desires, we still have a voice. But Rosa also theorizes on "uncontrollability" and wants us to seek resonance instead of control.

For Sandel and Rosa overly abstracted conceptions of autonomy are problematic. The major issue is not abstract autonomy itself, as they cautiously maintain "strong owning" and self-reflection and the subject interpreting itself. It is rather the overreaching of autonomy—its claims to contingency's rightful territory.

## 4. Rorty: Ossified Contingencies

In his *Contingency, Irony, and Solidarity*, Richard Rorty reflects even more poignantly than Sandel or Rosa on the role of contingency and autonomy.[10] His conception of the person is comprised with constant reference to contingency and irony—both of which are underwritten with a strong footing in self-creation and "willing". These latter concepts are highly celebrated in thinkers associated with authenticity. Indeed, throughout his work Rorty borrows heavily from Friedrich Nietzsche and many other 'authenticity' promoters.

Nietzsche's own project(s) embody Rorty's vision of philosophical endeavors. "He [Nietzache] saw self-knowledge as self-creation. The process of coming to know oneself, confronting one's contingency, tracking one's causes home, is identical with the process of inventing a new language—that is, of thinking up some new metaphors" (Rorty 1989, p. 27). We make sense of ourselves, others, and the world, through language or "vocabularies" which means not only actual words, but the concepts and norms that drive cohesion in interactions as well as a sense of individuality. The use of a vocabulary is what *creates* a self. Recognizing that the vocabularies inherited from others are completely contingent allows one to conquer the way others describe them and create their own language for self-description, which is ultimately self-creation. The new vocabulary is not, however, any less idiosyncratic than any other. The major difference is that the author of the new vocabulary can say "and thus I willed it". Though the process does not end here. In fact, it never ends.[11]

Fully acknowledging and appreciating contingency means accepting the endless task of continual redescription and re-creation of the self. It means being a poet. Someone

who does not "reiterat[e] a type, a copy or replica of something which has already been identified" (Rorty 1989, p. 28). Traditional views, Rorty thinks, see vocabularies as (1) being expressions of the self, which may be accurate or inaccurate, and (2) seek to find some external reference upon which to evaluate the veracity of said vocabularies. According to this way of thinking the "right" or "true" vocabulary should win out and unfamiliar vocabularies are eyed with distrustful skepticism. Rorty's poet or "ironist"—his ideal type of person—operates differently.

The ironist is someone who plays with vocabularies in order to explore new ways of creating their self. Understanding all vocabularies—one's own present vocabulary, the one(s) inherited, and even future ones—as contingent and idiosyncratic, all just as much "a product of time and chance" (Rorty 1989, p. 22) as any other, means submitting that they will never really reach the truth. This leads one to an ironic perspective. They realize that their own 'final' vocabulary is no 'truer' than any others and they allow themselves to be impressed by others.

The ironist "has radical and continuing doubts about the final vocabulary she currently uses" (Rorty 1989, p. 73) and frequently modifies her own. "[S]he does not think that her vocabulary is closer to reality than others, that it is in touch with a power not herself" (Rorty 1989, p. 73). Everything is treated "as a product of time and chance" and there is nothing left to worship or treat as even quasi-divine. In this private sphere the will is paramount. Rorty constantly references "Thus I willed it" as the determining factor in being a good ironist.

Willing one's own vocabulary defines the ironist. It is the mark of self-creation, of acknowledging and appreciating contingency, and being settled in inescapable idiosyncrasies.

> He is just doing the same thing which all ironists do—attempting autonomy. He is trying to get out from under inherited contingencies and make his own contingencies, get out from under an old final vocabulary and fashion one which will be all his own. The generic trait of ironists is that they do not hope to have their doubts about their final vocabularies settled by something larger than themselves. This means that their criterion for resolving doubts, their criterion of private perfection, is autonomy rather than affiliation to a power other than themselves. All any ironist can measure success against is the past—not by living up to it, but by redescribing it in his terms, thereby becoming able to say, "Thus I willed it". (Rorty 1989, p. 97)

Being able to say 'Thus I willed it' allows the ironist to escape being a mere type, copy, or replica of some already identified type. More importantly, it means fully accepting the role of contingency. No external power and not even correspondence to one's own self holds evaluative determinacy over one's vocabulary when 'Thus I willed it' is the measure of success.

Rorty's praise of autonomy is even more direct, and more frequent, than Sandel or Rosa's, and yet Rorty takes issue with the degree of abstractness which Sandel assigns to it (and would probably view Rosa similarly). In *Liberalism and the Limits of Justice* Sandel argues: "what is most essential to our personhood is not the ends we choose but our capacity to choose them. And this capacity is located in a self which must be prior to the ends it chooses" (Sandel 1982, p. 29). Rorty disagrees. As central as the will is to Rorty, it is not something that is somehow free from the contingencies that make up any other part of the self. He even goes so far as to affirm some version of the very "radically situated self" Sandel defends himself against. In direct response to Sandel Rorty writes,

> For if we use the vocabulary [of Dewey, Heidegger, Davidson, and Derrida], we shall be able to see moral progress as a history of making rather than finding, of poetic achievement by "radically situated" individuals and communities, rather than as the gradual unveiling, through the use of "reason", of "principles" or "rights" or "values". (Rorty 1990, p. 188)

While Sandel certainly does not subscribe to an Enlightenment-like view of moral progress, he does find that the "nature of the moral subject", is "in some sense necessary, non-contingent and prior to any particular experience" (Sandel 1982, p. 49). This is what Rorty takes issue with. In Sandel (and the same is true for Rosa) there is already too much in the way of "a core self which can look at, decide among, use, and express itself by means of, such beliefs and desires" (Rorty 1990, p. 10). Rorty actually takes up the position Sandel (and Rosa) cautiously skirts: "the view that human beings are centerless networks of beliefs and desires and that their vocabularies and opinions are determined by historical circumstance" (Rorty 1990, p. 189). Rorty is perfectly content with "a notion of the human self as a centerless web of historically conditioned beliefs and desires, the view that human beings are centerless networks of beliefs and desires and that their vocabularies and opinions are determined by historical circumstance" (Rorty 1990, p. 190). Autonomy is therefore subject to, or rather actually is, contingency—just as any other aspect of one's self or vocabulary. Autonomy is further praised for "replacing inherited [contingencies] with self-made contingencies" (Rorty 1989, p. 98), but not given a power outside of contingencies themselves.

### 5. Gray: Rejecting Narrative

John Gray is a staunch critic of autonomy, rationalism, and related notions of human progress. In his most recent book, *Feline Philosophy: Cats and the Meaning of Life* (2020), he takes on the notion of self-narrative.[12] He argues that humans get so caught up in narratives of the self that "the stories they have fashioned for themselves take over, and they spend their days trying to be the character they have invented" (Gray 2020, p. 105). Cats can be good role models for curing us of obsessing over creativity and narratives. "Cats do not plan their lives; they live them as they come. Humans cannot help making their lives into a story" (Gray 2020, p. 38). In the end, both actually live their lives 'by chance'. Telling stories and creating narratives of the self are human ways to find reasons for things, to impose autonomy, and ultimately divert ourselves from the determining force of contingency.

Like Rorty, Gray thinks most philosophical inquiry fails because philosophers usually "imagine life can be ordered by human reason". However, "[o]ur lives are shaped by chance and our emotions by the body". But "[m]uch of human life—and much of philosophy—is an attempt to divert ourselves from this fact" (Gray 2020, p. 31). In other words, many philosophical descriptions seek to explain away (at least some degree of) contingency. Drawing on Nietzsche and Spinoza, Gray argues that "what we think are our choices are [actually] the result of complex causes operating in our organism" (Gray 2020, p. 52). Accordingly,

> Our thoughts and decisions are not separate from our bodies, which function independently of what we take to be our conscious mind and will. The experience of deliberating and deciding to pursue an option is a by-product of our conflicting desires. Free will is the sensation of not knowing what you are going to do. (Gray 2020, pp. 52–53)

Gray sees a lot of overlap between this understanding, which he mainly attributes to Spinoza, and traditional Daoist philosophy. Both, he says, "set limits on the human will". (Gray 2020, p. 53) In this way Gray goes even further than Rorty in minimalizing the degree of autonomy and significance it may have in his conception of the person.

Already freed from tying life to an external social order[13] individualist thinkers such as Sandel and Rosa locate the determining power of individual and collective identities in some facet of the self. A force which is prior to, and independent of, specific aims, preferences, desire, and the like. Rorty rejects this self. There is nothing prior or independent. This does not mean, however, that we must fall back on "reiterating a type, a copy or replica of something which has already been identified" (Rorty 1989, p. 28). In other words, we do not need to go back to a social-roles-based view of ourselves, others, and society. We escape this external power not by finding an intrinsic self as a replacement, but by constantly

creating, as contingencies ourselves, new contingencies. John Gray comes down harshly on these narratives. They are both inaccurate and not useful.[14]

Gray couples Rorty with Jean-Paul Sartre and others commonly associated with authenticity. He summarizes their positions: "Human beings create themselves, they say. Unlike other animals, they can choose to be whatever they want to be" (Gray 2020, p. 106). Contrary to this, Gray thinks human beings have a nature[15] that is not entirely created or subject to constant re-creation. That nature means "the good life for each of us is not chosen but found" (Gray 2020, p. 107). Here, he borrows again from a "[D]aoist belief that we must follow the way within us" (Gray 2020, p. 107). The "I" that follows, or should follow, the "way" within us is not a unified indivisible subject. Sounding like Nietzsche, and to some extent Rorty as well, Gray writes: "There is no self that is more or less self-aware, only a jumble of experiences that are more or less coherent". He goes on to reject the foundation for any "Thus I willed it" and couples it with a praise of cats: "We pass through our lives fragmented and disconnected, appearing and reappearing like ghosts, while cats that have no self are always themselves" (Gray 2020, p. 66). To be like a cat means to be settled in being fragmented and disconnected. It means not telling oneself a story about oneself nor engaging in attempts at autonomy or creation.

One way to describe what Gray is getting at is "spontaneity". Many interpretations of this term are, however, close to the "blowing in the wind" views of the self Sandel and Rosa adamantly reject. Having freed the self from being almost entirely based on social roles a 'self' was given determinative power. If this self is not an antecedent self (Sandel and Rosa), then it should at least be endowed with the power to will and create (Rorty). Gray's rejection is a critique of the supposed dichotomy between autonomy and "radically situated" relativistic "voluptuaries". The last sentence of *Feline Philosophy* reads "The meaning of life is a touch, a scent, which comes by chance and is gone before you know it" (Gray 2020, p. 111). Gray's acceptance of contingency does not seek a way out for the self. It is simply this acceptance and nothing more. Taking life as it comes means getting past fixed notions of how things should be and of what it means for life to be "perfect"; Gray writes:

> One burden we can give up is the idea that there could be a perfect life. It is not that our lives are inevitably perfect. They are richer than any idea of perfection. The good life is not a life you might have led or may yet lead, but the life you already have. Here, cats can be our teachers, for they do not miss the lives they have not lived. (Gray 2020, p. 108)

Gray thinks humans are rarely spontaneous in the sense of being completely absorbed in what they are doing, i.e., accepting life as perfect and not missing what is not the case. Borrowing from Zen, and ultimately Daoism, Gray says this is having "no-mind"—which is not the same as being mindless. It means, most simply, being without pre-conceived notions of how things ought to be.

To summarize: In these discussions contingency and autonomy are in a see-saw relationship. Increasing one results in a corresponding decrease in the other. Gray's position is the most extreme. Much of *Feline Philosophy* is an outright rejection of contemporary conceptions of autonomy. Even Richard Rorty, largely taken as quite radical in his own right, is rejected for having too strong a notion of self.

For over two decades Gray has been drawing heavily on Daoist (Taoist) sources. Although he mainly references the *Daodejing* (*Tao-Te-Ch'ing*) or *Laozi* (*Lao-Tzu*), his arguments in *Feline Philosophy* are actually much closer to what is found in the second most famous Daoist classic, the *Zhuangzi* (*Chuang-Tzu*).[16] Here much of what Gray hints at in terms of accepting life, appreciating contingency, and not having a self, are fleshed out in more detail.

## 6. Daoism: Alternative Approaches

Classical Daoist philosophy does not reflect on the relationship between contingency and autonomy in the same way as the abovementioned authors. While many of the

concerns central in Sandel, Rosa, Rorty, and Gray are echoed in the *Zhuangzi* and *Laozi*, the latter were written in times when conceptions of the self, society, and politics were vastly different. For this reason, this paper is not meant to argue that Daoist philosophy will replace these thinkers, or somehow offer better alternatives. The resources sketched below allow us to approach well-worn patterns of thought from perspectives which can allow previously unreflected upon issues to bubble to the surface.

Sandel's worry about the focus on merit in contemporary American society—the way it makes winners "inhale too deeply of their success", and convinces losers that they really are losers—is echoed in the *Zhuangzi*. The text discusses self-understanding in terms of recognizing the contingency of merit and how it informs people's conceptions of themselves (and others):

> And he whose knowledge is sufficient to fill some one post, or whose deeds meet the needs of some one village, or whose personal virtues please some one ruler, or who is able to prove himself in a single country, sees himself in just the same way. (1.3)[17]

The "some one" is a recognition of contingency. A person's knowledge, for example, does not "rightfully" fill the post, it is just a matter of luck. The person's ability just happens to suit what just happens to be desired. Any type of merit, including knowledge, deeds, or virtues are not deserving of anything in and of themselves. Their positive or negative evaluations are entirely subject to contingent factors. Qualities praised at one historical moment, in one culture, or even in a specific generation, region, company, or friend group can hugely vary. Fulfilling the requirements of a post, behaving in a way deemed desirable by some people, having some virtues or talents that certain people happen to like, or even being elected president, are all due to completely conditional circumstances. Some other job, other people, another boss, or different voters might completely reject the same person who was previously so successful. It is therefore unnecessary to take social praises or condemnations as defining of oneself or of anyone else.

The criticism of merit in the *Zhuangzi* cuts deeper (and along different lines) than with Sandel. If Sandel thinks we shouldn't change our system of merit to make a better meritocracy because it will never foster healthy conceptions of people and therefore asks to entirely reconsider its very foundations and develop a politics of the common good, the *Zhuangzi* thinks we shouldn't change our politics to try and foster better self-understandings, but rather maintain a critical distance between our social self and the way we see ourselves. This Daoist split means being fully involved in engagements with others and with conventions but allows that they are not taken so seriously as to be damaging to the self internally. It avoids anxieties caused by being a failure or a success. We can temporarily be attached while simultaneously maintaining a critical distance.

The *Zhuangzi* often pits itself against the more merit-rewarding Confucius to emphasize its appreciation of contingency. In one such instance the *Zhuangzi* tells a fictional story of one of Confucius' students, Zigong, coming across an old man wobbling to his garden with heavy pales of water. Zigong, a great entrepreneur, suggests he make a water pulley device. It would be far easier to water the few plants he has and he could even expand his operation. The gardener is angered. Then he smiles and laughs it off. He says: "having mechanized instruments necessarily means engaging in instrumentalized activities, engaging in these activities necessarily means having a mechanical heart-mind" (12.11). Upon hearing Zigong is a disciple of Confucius the gardener rhetorically asks if he is the master who is trying to "sell his name to the world" and "forgets his own spirit" in the process. The gardener is clearly not interested in the pulley system at all.[18] Zigong leaves, and after a time passes realizes that for someone who does not engage in mechanical or instrumentalized thinking "accomplishment, profit, machinations, and skill" are all forgotten. He describes the gardener to Confucius as someone who does not let other people's praise or condemnations bother him. This is, paradoxically, idealized by the two along with being able "throw your life in with the ordinary people, moving together with them and never knowing where you're going".

One who is able to "throw their life in with ordinary people" accepts the way they are viewed by others without internalizing it. They ride conventions without getting caught up in them. "Never knowing where they are going" expresses not taking an instrumental approach to their own identity, merit, name, or reputation. This is contrasted with "selling one's name", whereby one cultivates their name and reputation according to what is conventionally praised. The goal is to get the best price—gain as much as possible. This attitude easily results in overly attaching to developing a notion of self intertwined with social appraisals. In other words, taking oneself and social conventions too seriously and not appreciating the pervasiveness of contingency.

Cultivating a name to sell is like growing vegetables to hawk. It involves expanding otherwise natural practices where one is not consciously engaged in mechanistic thinking to instrumentalized calculations. Rosa speaks as if appropriating, dominating, and controlling the world influences how we think of ourselves, others, and our relationship with the world. His is in the privileged position of an observer, and he describes subjects as being able to take on such a position themselves. The *Zhuangzi* is more extreme. It argues that mechanizing even some of our behaviors result in our own heart and mind becoming mechanical. In another place the text warns of not "being thinged by things" (11.5) when using them. In other words, treating oneself like a tool by concentrating on productivity, efficiency, and what is "gained" through our actions leads to one actually becoming such a tool.

The later Daoist Guo Xiang uses "vanishing into one another" to discuss contingency in the *Zhuangzi*. Each thing constantly transforms through interactions with everything else in its environment—and these things are, in turn, all transforming themselves. Everyday examples include how the weather influences one's dress and mood, which, in turn, influences how one interacts with the world and others around them. Even a simple look from a stranger, a conversation with one's partner, or some untimely bird shit can have a huge impact on a person's day, week, or life. According to this model contingency and autonomy are no longer significantly distinguished from one another. Whatever one does is always only ever manifest within concrete contingencies. Any degree of reflection or evaluation we make upon ourselves is derived from how we have grown up within contingencies. No force from the outside is imagined as producing any privileged position.

Rosa's resonance solution calls for one to be open to the world and its contingencies. However, from the privileged position of an observer instrumentalization is seen as happening to someone, it is not the person themselves. The subject, like Rosa, can step outside of contingencies to analyze them and their effect. The *Zhuangzi* disagrees. When people engage in instrumentalizing activities, they themselves become instrumentalized. Rejecting mechanistic behaviors, feelings, and thinking means letting go of the idea of selling one's name and instead going along with conventions, but not being strongly attached to them. The *Zhuangzi* offers non-doing, self-so, and play as alternatives. Resonance might be the result, but it is not something to be sought after, nor can it be (as Rosa himself argues, albeit somewhat paradoxically).

The thoroughgoing nature of contingency in the *Zhuangzi* reminds one of Rorty's understanding. Although their "vocabularies" are distinct they overlap in finding the person entirely subject to historical and cultural circumstance. For Rorty, the way "out"—in the sense of staking some ownership—is through the power of willing—even though this too does not escape contingency. The will for Rorty simply allows one to be creative and encourages reflection. The *Zhuangzi* sides with Gray in not looking for any way out. There is no explicit rejection of will or creativity in the Daoist texts, they simply do not recommend them. Either would be too purposeful and degrade one's more "self-so" nature.

Gray's provocative work is largely directed against prominent understandings of progress, reason, theism, and autonomy. The alternative he offers is mainly through negation. The *Zhuangzi* (and *Laozi*) provides resources for exploring what letting go of these notions and appreciating contingencies might look like with more concrete discussions.

It even adds playfulness, which is surprisingly missing in Gray's *Feline Philosophy*, as a central element.

### 7. Feeling Autonomous

In their *Genuine Pretending: The Philosophy of the Zhuangzi*, Hans-Georg Moeller and Paul D'Ambrosio ([Moeller and D'Ambrosio 2017](#)) explore the Daoist ideal of adapting to social conventions without becoming attached to them. Their work expressly addresses how we utilize our sense of autonomy while recognizing it as always already embedded within contingency.

Moeller and D'Ambrosio explain genuineness in the *Zhuangzi* as fully accepting the purchase of internal and external commitments in a descriptive sense. Coming to terms with the relative determinacy of various psychological elements, biological processes, and social expectations, including roles, esteem, and reputation, allows one to fully perform them. One does so through recognizing that they themselves, just as much as anyone else and as social structures in general, are essentially contingent. We experience (the feeling of) autonomy, but that does not mean it exists outside of contingency. Evaluative adjustments and reflection are done from within the subject's own idiosyncratic contingencies and through their contingent interactions to other contingencies. According to Moeller and D'Ambrosio this *Zhuangzi*ian view allows us to understand how people can fully adopt their roles, relate to others and the world, and understand their own "self". The fact that we feel as if "we" do "adopt" roles speaks to our experience of a fundamental disconnect between concrete contingencies. In the *Zhuangzi* our experience of this distinction is referred to as "play", that is, the ability to shift between identifying with our various contingencies without getting overly attached to them.

The reflective space for making evaluations, for envisioning change, and for the phenomenal experience of "autonomy" comes from our not actually being what our roles or any other external expectations call us to be. Our contingencies make us. They are multifarious and incongruent, and so, therefore, are we. Accordingly, even if we internalize our contingencies we can only do so as fractures. The "playful" element of the *Zhuangzi* notices that our shifting, multifarious, and incongruent contingencies entail a fundamental distance to any one in particular. This is what Moeller and D'Ambrosio refer to as "pretending". Instead of attempting to unifying them into a singular identity, hypothesizing an underlying will, or positing any abstract or fixed self worthy of merit, or ability to transcend contingencies and evaluate them externally, the *Zhuangzi* presents us with the constant oscillation between being genuine and playfulness, in other words, "genuine pretending".

From the perspective of "contingency-autonomy", pure contingency is paramount to neglecting our experience of choice and of self. If we are only contingencies there is no space to explain the experience "*I* have made that *choice*". We would be, at worst, what Rosa describes as "pure voluptuaries" always giving into our contingencies (which critics tend to associate with our most base desires), and at best we a "merely situated self". The genuine pretending model "solves" this problem because it exists in an entirely different framework. It lays out how the *Zhuangzi* does the same work of addressing these issues and experiences without reference to an external autonomy. We can say indeed say "*I* have made that *choice*" from within the realm of contingency.

Genuine pretending adopts all of the same contingencies from the "contingency-autonomy" model. We are constituted by our social environments, biological processes, psychological elements, and all sorts of interactions with others and with the environment. These are given, uncontrollable, contingencies. The genuine pretending model emphasizes that how they constitute us, the demands they make, and changes they produce are wildly multifarious and incongruent. In adopting whatever one at whatever time we can be genuine. However, we never stay for long. Our contingencies are in constant motion. Their product, or in the space where they rub against one another when they are no longer harmoniously meshed, is our experience of a self. In other words, the self is precisely the experience of discordance between various contingencies. Any choice or decision is also

a presentation of competing contingencies. When we make a choice, which the *Zhuangzi* says is often "self-picked" or happens in a "self-so" manner, it is not a non-contingency constituted "self" that has made the choosing. We may strongly feel that some power or self has decided to choose this over that. It is, however, not an autonomous self who has chosen. Certain contingencies win out over others, and the experience of the initial tension and resulting acquiesces is "*I* have made that *choice*". Or, we can think of this as some stronger or more cultivated contingency winning out over others. In this way *we* really have *made a choice*, but this is not something worthy of attachment or ownership, it simply is the way we experience ourselves.

In sum, according to the contingency-autonomy model I really do make a choice. The feeling we have is descriptive of a force outside of contingency. The genuine pretending model differs. It says that we *experience* "I made a choice" but this does not mean there is an stable or absolute "I" outside of contingencies.

## 8. Conclusions

Implicit in the works of Michael Sandel, Hartmut Rosa, Richard Rorty, and John Gray is the autonomy-contingency binary. They all suppose that the more autonomy is granted in one sphere, the less contingency is recognized—and vice versa. Viewing things in terms of this see-saw relationship they each call for fuller recognition of the reach of contingency in their respective spheres of concentration. With the exception of Gray, they all, like many contemporary Western thinkers, hold onto strong versions of autonomy. The individual can never fully be given over to contingency for that would potentially erase everything from social and political steering to moral and personal responsibility. This binary and the harsh lean on autonomy figure prominently in various concrete areas of contemporary life as well. Obama's 2012 "You Didn't Build That" speech sparked a huge debate about responsibility, autonomy, and contingency. It was widely discussed by news anchors, politicians, and people from nearly every walk of life. In religion, in choosing a college, a bedtime snack, a partner or career, in mental health and addiction, and all sorts of other existential, socio-political, and spiritual arenas autonomy and contingency are juxtaposed. (The "serenity prayer" is a concise example.)

In Daoist texts we do not see this familiar binary. Concepts related to autonomy are relegated to the realm of contingency. Persons are their contingencies, and yet no one contingency is individually exhaustive. It is the gambit of our contingencies as well as their interactions that constitute our experience of "self", "choice", (and "autonomy"). The sense of self and experience of making a choice are every bit as real as we experience them, they simply are neither the indicator nor product of a power outside of contingency.

The implications of this alternative model are far-reaching. For example, responsibility, in a general and also moral sense, can be seen differently. In early China punishments and well as honors could easily extend well beyond the individual who "earned" them. Entire clans could be executed or elevated based on the idea that no single person is responsible for who they are. Persons are cultivated through their interactions with others. So who one becomes, what they are like, and what they do, is always "shared". The same framework exists in China today. Crimes and achievements are often thought of in relatively broad contexts.[19] Additionally, The People's Republic of China is often characterized as a meritocracy—and China has a long history of putting a strong emphasis on merit in social and political spheres. The problems Sandel brings up are, however, conspicuously lessened in a culture where individuals are not thought of as indivisible.[20]

The Daoist model does not "solve" problems outlined by Sandel, Rosa, Rorty, or Gray. Chinese resources do, however, offer a model for making otherwise familiar ideas strange, for taking what is often overlooked and reassessing it. Re-envisioning the relationship between contingency and autonomy allows a complete rethinking of meritocratic hubris, resonance, our relationship to our vocabularies and will, and even how we look for alternatives to narratives of the self.

**Funding:** This research was funded by Fundamental Research Funds for the Central Universities (No. 2018ECNU-QKT010).

**Institutional Review Board Statement:** Not applicable.

**Informed Consent Statement:** Not applicable.

**Data Availability Statement:** Not applicable.

**Conflicts of Interest:** The author declares no conflict of interest.

## Notes

[1] Sandel's book is somewhat of an exception, with a COVID-19 discussion added in before the original, and much better, introduction.

[2] In this paper 'self' will generally refer to how a person understands themselves. More specific notions will be used at times, for instance in discussing Rorty and Daoism.

[3] The connotations of this word are perhaps reflective of Aristotle's own feelings about the influence context has on a person.

[4] A form of this debate has been argued publicly in a wider (though much less academic forum) after Obama's 'you didn't build that' speech in 2012. (cf. https://en.wikipedia.org/wiki/You_didn%27t_build_that, accessed on 28 August 2021).

[5] Of course we still need to cultivate our skills. Whether or not motivation makes it into the category of personal responsibility is still up for debate, though many might not be happy with where the science is leaning in terms of finding evidence that it has strong biological foundations. (cf. Deckers 2018) There are also many environmental factors to be taken into consideration. Sandel's argument, however, does not rely on hashing out these details.

[6] Everything mentioned about Rosa up to this point is greatly expanded upon in *Social Acceleration* (Rosa 2013), where he provides a penetrating analysis of acceleration in contemporary society.

[7] The quotes within these quotes are from Rosa 2016.

[8] A separate but closely related issues is Theodor Adorno's concept of 'identity thinking' which Rosa also discusses (Rosa 2020, pp. 97–98).

[9] Rosa imagines silly knocks against his work '*What are you writing a book about?* [ . . . ] *That sometimes chance is determinative?*' (Rosa 2020, p. 100).

[10] Solidarity, which is also a major topic for Rorty, is essential to his political views, and is not directly relevant here.

[11] Rorty writes: '"Thus I willed it", it will remain true that this willing will always be a project rather than a result, a project which life does not last long enough to complete' (Rorty 1989, p. 40).

[12] Gray has made similar arguments in earlier works, especially in his *Straw Dogs* (Gray 2002) and *The Soul of a Marionet* (Gray 2015).

[13] The corresponding view of the self or identity in pre-individualistic model was largely based on social roles. Nearly everything about a person, including the way they understood themselves, others, and the world, was based on their position in society. When this breaks down, the typical response has been to locate the determining power of individual and collective identities in the self.

[14] They are not "useless" they do serve some purposes. But we would be better off without them, with those purposes left unserved.

[15] In other words, he is against Sartre's famous "existence before essence".

[16] Gray also draws heavily on Zen, which is the Japanese adaptation of Chan Buddhism. The latter was developed in China, and is considered, by many Buddhologists, considered at least as close, if not closer, to Daoism. Most of the vocabulary and conceptual reconfigurations that characterize Chan (and Zen) are direct adaptation from the *Zhuangzi*.

[17] Translations not otherwise noted are my own. The chapter verse numbering refers to ctext.org.

[18] Voltaire's *Il faut cultivar notre jardin* or "cultivating one's own garden" is a happy coincidence of terms. The Daoist gardener is saying, literally and metaphorically, more or less the same thing.

[19] There is, of course, some overlap here. With particularly bad crimes or morally disturbing behavior we often ask 'where was the family' however, relatively speaking, more responsibility is assigned to the family, friends, and teachers in China than in, for example, the U.S. today.

[20] Compare Sandel's discussion of meritocracy in American with Daniel Bell's analysis of meritocracy in China (Bell 2016).

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
