# Peer review of "Daoist Reflections on the See-Saw of Contingency and Autonomy: The Laozi and Zhuangzi in Dialogue with Sandel, Rosa, Rorty, Gray"

_religions, doi:10.3390/rel13100972_

Round 1
Reviewer 1 Report
This is a brilliant paper, expressing a powerful and distinctive point of view on the question of contingency and autonomy, well contextualized within contemporary debates on these subjects, and with a broad comparative vision that clarifies the issue in novel ways due to well-chosen juxtapositions and exposition. It makes a significant contribution to both defining and advancing a conversation with immense present relevance both to philosophy and to broader social and cultural issues. I recommend it be published as is.
Author Response
Thank you.
Reviewer 2 Report
My only concern is the last paragraph where a judgment is rendered regarding the fact that modern thinkers do a much better job than Zhuangzi in solving the issues they raise, though it is not at all clear that Zhuangzi is trying to solve any problems, so it might be a little unfair, and you don't seem to have really argued for that in the paper. It might not be necessary or helpful in the conclusion.
Author Response
I agree with the point about the last paragraph.